# Equivariant Graph Self-Attention Transformer for Learning Higher-Order Interactions in 3D Molecular Structures

## Abstract

Despite their considerable success in multiple fields, studying 3D molecular structures of varying sizes presents a significant challenge in machine learning, particularly in drug discovery, as existing methods often struggle to accurately capture complex geometric relationships and tend to be less effective at generalizing across diverse molecular environments. To address these limitations, we propose a novel Equivariant Graph Self-Attention Transformer, namely EG-SAT, which effectively leverages both geometric and relational features of molecular data while maintaining equivariance under Euclidean transformations. This approach enables the model to capture molecular geometry through higher-order representations, enhancing its ability to understand intricate spatial relationships and atomic interactions. By effectively modeling the radial and angular distributions of neighboring atoms within a specified cutoff distance using Atom-Centered Symmetry Functions (ACSFs), EG-SAT leads to a more nuanced and comprehensive understanding of molecular interactions. We validate our model on the QM9 and MD17 datasets, demonstrating that EG-SAT achieves state-of-the-art performance in predicting most quantum mechanical properties, thus showcasing its effectiveness and robustness in this domain.

## 1 Introduction

Geometric deep learning has gained prominence as a powerful approach that leverages the inherent symmetries of specific learning tasks by embedding geometric priors (Satorras et al., 2021; Bronstein et al., 2021; Chmiela et al., 2018). By embedding these priors, models are endowed with a significant inductive bias, which narrows the scope of learnable functions and leads to enhanced performance. Classic examples include Convolutional Neural Networks (CNNs) (LeCun et al., 1995), which are designed to maintain in each layer equivariance to translations, and Graph Neural Networks (GNNs) (Kipf & Welling, 2016), which are inherently invariant to node permutations. These models, by exploiting symmetry groups, have been instrumental across a range of applications.

Among these, GNNs have seen extensive success in modeling molecular structures, from small molecules for quantum chemistry predictions (Gilmer et al., 2017; Schütt et al., 2017) to larger macromolecular structures like proteins (Ingraham et al., 2019; Fout et al., 2017). Their effectiveness lies in their ability to represent molecules as graphs, where atoms are treated as nodes and the edges are defined by either chemical bonds or spatial proximity. GNNs commonly capture molecular geometry using rotationally invariant features such as pairwise distances to model local interactions. However, while this approach has been widely adopted, it lacks the ability to encode directional information, which is critical for fully capturing the spatial arrangement and interactions between atoms.

To address these shortcomings, the inclusion of more comprehensive geometric features is necessary. GNNs (Kipf & Welling, 2016; Hou et al., 2020) excel at representing atomic systems due to their natural fit with 3D graph representations, where atoms are defined by their Cartesian coordinates. Incorporating 3D molecular data, including bond lengths and angles, is crucial for improving model performance (Schütt et al., 2017; Chen et al., 2019; Gasteiger et al., 2020b). However, purely rotationally invariant representations can blur distinctions between different structures, causing the

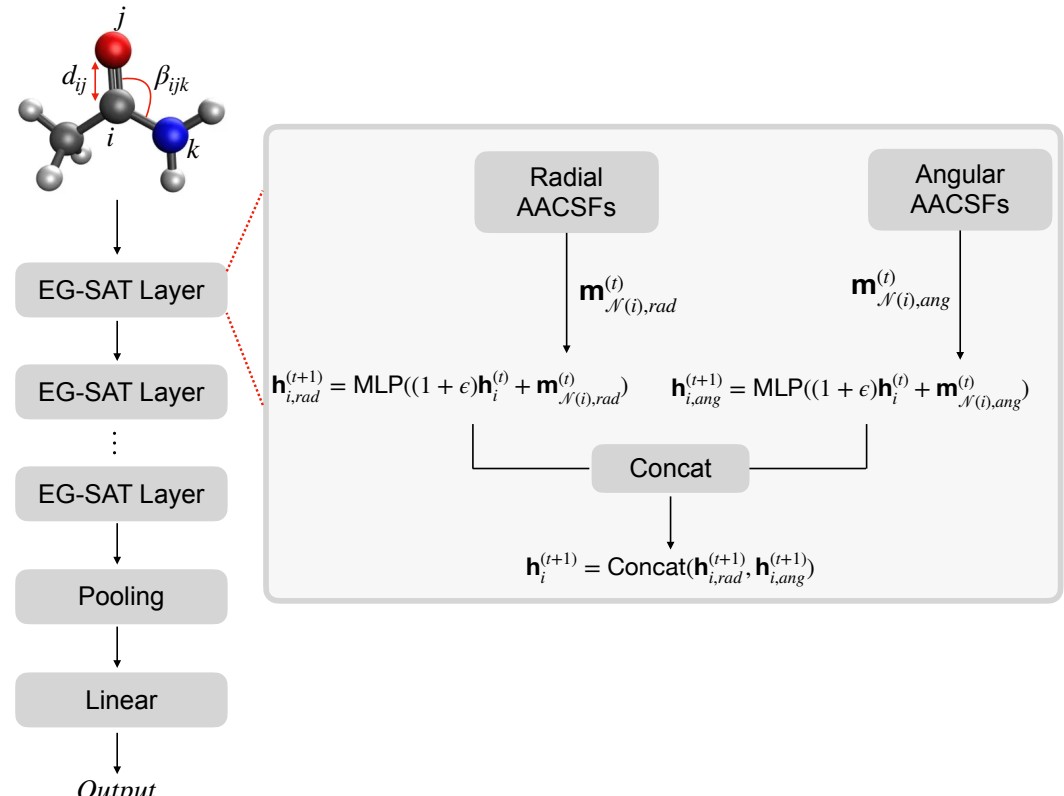

Figure 1: A high-level overview of the EG-SAT model architecture. We subtly account for the chemical composition of the environment by incorporating element-specific attention mechanisms based on Radial AACSFs and Angular AACSFs, as shows in Figure 2. We derive the final result for the radial attention mechanism, denoted as $\mathbf{h}_{i,rad}^{(t+1)}$. In the same way, we calculate the outcome for the angular attention, $\mathbf{h}_{i,ang}^{(t+1)}$. Finally, we combine these two results to form the final node representation, $\mathbf{h}_i^{(t+1)}$.

model to treat distinct atomic configurations as identical (Miller et al., 2020; Schütt et al., 2021). When only pairwise distances are used as edge features, critical information such as bond angles can be lost, especially for properties that are angle-dependent, like optical absorption (Hsu et al., 2022). Even when angular information is included to address triplet interactions, there are still challenges in distinguishing molecular chains with equivalent bond angles as the structure rotates, altering dihedral angles.

Equivariant neural networks offer a promising solution to this problem by capturing the inherent symmetries in molecular data (Weiler et al., 2018). These networks, particularly those based on irreducible representations (irreps) (Batzner et al., 2022; Thomas et al., 2018; Musaelian et al., 2023), employ spherical harmonics to generate higher-order representations that account for these symmetries, resulting in improved accuracy in predicting molecular properties. Recent advancements in this area have integrated attention mechanisms to further enhance performance (Fuchs et al., 2020; Liao & Smidt, 2023). However, irreps come with the downside of high computational costs, especially when dealing with higher-order transformations. In contrast, more efficient equivariant vector representations that operate directly in 3D Cartesian space have shown to achieve comparable state-of-the-art performance across various tasks with significantly lower computational overhead (Schütt et al., 2021; Thölke & De Fabritiis, 2022a).

To overcome the limitations of distance-based representations and improve scalability, it is essential to integrate angular information, which encodes bond angles and spatial relationships between atoms, offering critical directional cues. By incorporating such geometric features into GNN architectures,

models can better capture the local geometry of molecules, leading to enhanced predictive performance (Gasteiger et al., 2020b). However, GNNs must also be able to handle larger molecular structures with thousands of atoms and process their 3D geometry in a manner that is independent of orientation and position. Enforcing equivariance under transformations corresponding to specific symmetry groups ensures that the model can consistently analyze molecular data across different orientations (Bronstein et al., 2021; Battaglia et al., 2018), bridging the gap between local interactions and global structural consistency.

Despite these advances, scaling models that incorporate geometric information remains computationally demanding, particularly when dealing with large molecular systems and a diverse range of chemical species. One common method for capturing geometric features is through Atom-Centered Symmetry Functions (ACSFs), which encode radial and angular distributions around each atom. However, ACSFs face scalability issues, as the number of symmetry functions grows rapidly with the number of chemical elements in a molecule, resulting in large descriptor vectors and increased computational costs in high-throughput settings, such as molecular dynamics simulations (Gastegger et al., 2018).

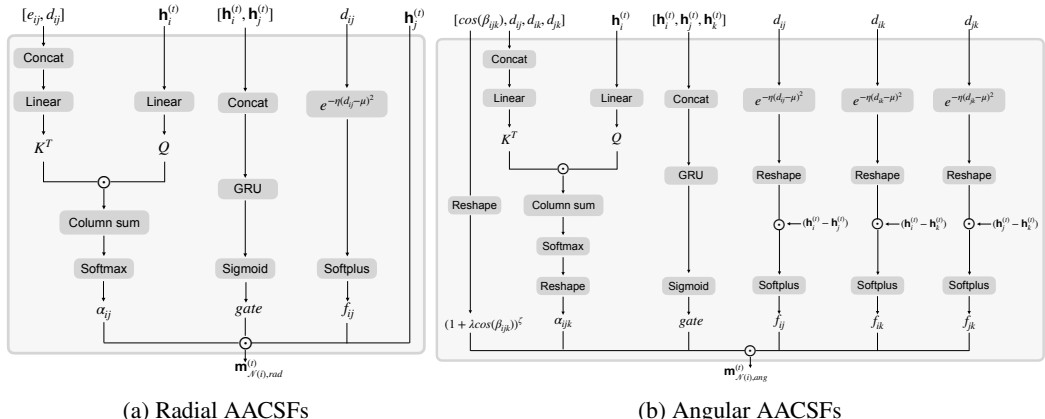

(a) Radial AACSFs          (b) Angular AACSFs

Figure 2: (a) Radial AACSFs adjusts contributions from atom pairs based on their distances and features, while the gating mechanism provides flexibility and the Gaussian radial decay emphasizes interactions with nearby atoms; (b) Angular AACSFs assigns varying importance to angular features, while the gating mechanism manages information flow among atom triplets.

In this paper, we propose a novel approach that enhances traditional ACSFs by incorporating attention mechanisms. Our method integrates both angular and radial information while addressing the scalability challenges associated with ACSFs. This approach maintains computational efficiency while improving predictive performance in GNNs for molecular applications. We rigorously validate our model on benchmark datasets, demonstrating that it significantly enhances the understanding of molecular structures and shows promise for applications in drug discovery and material science.

The main contributins of this work are follows:

- We introduce a new class of Attention-based Atom-Centered Symmetry Functions (AACSFs), which maintain roto-translational invariance while improving on traditional ACSFs by incorporating element-specific attention mechanisms.

- We develop a novel Equivariant Graph Self-Attention Transformer (EG-SAT), which is theoretically analyzed to ensure it respects the necessary symmetry properties of 3D molecular structures.

- Our model addresses scalability issues of ACSFs by using attention-based mechanisms, allowing efficient representation of diverse chemical elements without exponentially increasing the number of symmetry functions.

- We propose a new gating mechanism that modulates the contributions of atomic pairs and triplets, while the attention mechanism dynamically adjusts the weighting of angular features. This enables the model to capture intricate geometric relationships between atoms more effectively.

## 2 BACKGROUND

In this section, we formalize the underlying geometric graph representation and introduce key concepts related to symmetry, equivariance, and the atom-centered symmetry functions (ACSFs) that play a vital role in molecular graph neural networks.

Let $G = (V, E, \mathbf{A}, \mathbf{H}, \vec{\mathbf{F}}, \vec{\mathbf{X}})$ be an undirected geometric graph, where $V$ is a set of nodes, $E \subseteq V \times V$ is the set of edges. The matrix $\mathbf{A} \in \mathbb{R}^{n \times n}$ defines the adjacency relationships between nodes, $\mathbf{H} \in \mathbb{R}^{n \times z}$ represents scalar node features, $\vec{\mathbf{F}} \in \mathbb{R}^{n \times d}$ is the matrix of geometric features, and $\vec{\mathbf{X}} \in \mathbb{R}^{n \times 3}$ contains the spatial coordinates of the nodes.

### 2.1 SYMMETRY AND GROUP REPRESENTATIONS

A critical aspect of geometric graphs is their symmetry properties. These are captured by the group of symmetries $\Pi$, which preserves the structural relationships in the graph. Formally, an n-dimensional representation of a group $\Pi$ is a mapping $\rho : \Pi \to \mathbb{R}^{n \times n}$, where each group element $\pi \in \Pi$ is associated with an invertible matrix $\rho(\pi)$. This mapping satisfies the condition $\rho(\pi\omega) = \rho(\pi)\rho(\omega)$ for all $\pi, \omega \in \Pi$. When $\rho(\pi)$ is orthogonal for all $\pi \in \Pi$, the representation is called orthogonal.

Group representations enable us to define two key properties:

- **Invariance**: A function $f : X \to Y$ is $\Pi$-invariant if $f(\rho(\pi)x) = f(x)$ for all $\pi \in \Pi$, meaning the output remains unchanged under the group action.
- **Equivariance**: A function $f : X \to Y$ is $\Pi$-equivariant if $f(\rho(\pi)x) = \rho(\pi)f(x)$ for all $\pi \in \Pi$, meaning the group action on the input results in a corresponding transformation of the output.

### 2.2 GROUP ACTIONS ON GEOMETRIC GRAPHS

In geometric graphs, symmetry operations can transform the geometric attributes of the nodes and edges as follows:

- **Permutations**: Given a permutation matrix $\mathbf{P}_\sigma$, a permutation of the graph is defined by $\mathbf{P}_\sigma \mathcal{G} = (\mathbf{P}_\sigma \mathbf{A} \mathbf{P}_\sigma^T, \mathbf{P}_\sigma \mathbf{H}, \mathbf{P}_\sigma \vec{\mathbf{F}}, \mathbf{P}_\sigma \vec{\mathbf{X}})$.
- **Orthogonal Transformations**: Let $\mathbf{Q} \in \Pi$ be an orthogonal transformation that acts on the geometric features and coordinates as $\mathbf{Q}\vec{\mathbf{F}}$ and $\mathbf{Q}\vec{\mathbf{X}}$, respectively.
- **Translations**: For a translation vector $\vec{\mathbf{t}} \in \mathbf{T}$, the node coordinates are translated as $\vec{\mathbf{x}}_i + \vec{\mathbf{t}}$ for all nodes $i$.

These transformations capture the symmetries in the geometric structure of the graph, allowing models to exploit equivariance in learning representations.

### 2.3 EQUIVARIANCE IN 3D SPACE

In the context of 3D molecular graphs, the relevant symmetry group is the Euclidean group SE(3), which includes rotations and translations. This group can be mathematically represented through its action on points in three-dimensional space. For any 3D orthogonal matrix $\mathbf{Q} \in \mathbb{R}^{3 \times 3}$ and a translation vector $\vec{\mathbf{z}} \in \mathbb{R}^3$, the group action $\rho(.)$ on $\pi \in SE(3)$ is defined as:

$$\rho(\pi)\vec{\mathbf{x}} = \mathbf{Q}\vec{\mathbf{x}} + \vec{\mathbf{z}}, \tag{1}$$

where $\vec{\mathbf{x}} \in \mathbb{R}^3$ represents the 3D coordinate vector. This formulation captures transformations induced by rotations, reflections, and translations, establishing a foundational understanding of how molecular structures can be manipulated in 3D space. Within this context, the subgroup SO(3) specifically represents the rotational symmetries, isolating the behavior of molecular graphs under rotation alone.

To further clarify the structure of these groups, we consider the orthogonal group O(3), which includes all 3D orthogonal matrices. It is defined as: $O(3) = \{\mathbf{Q} \in \mathbb{R}^{3 \times 3} | \mathbf{Q}^T \mathbf{Q} = \mathbf{Q}\mathbf{Q}^T = \mathbf{I}, det(\mathbf{Q}) = \pm 1\}$. This group encompasses both rotations and reflections, highlighting the full range of orthogonal

transformations in three dimensions. Notably, the special orthogonal group SO(3) is a subset of O(3) that is restricted to those orthogonal matrices with a determinant of +1. This restriction is crucial, as it ensures that the transformations in SO(3) represent pure rotations, which are particularly relevant in the context of molecular configurations where orientation is a key factor.

## 2.4 ATOM-CENTERED SYMMETRY FUNCTIONS (ACSFs)

ACSFs (Gastegger et al., 2018) are an essential tool for capturing the local geometric environment of atoms in molecular systems. They are designed to ensure invariance to symmetries in atomic arrangements, providing a mathematically rigorous representation of the molecular structure.

The radial ACSF encodes the distribution of neighboring atoms around a central atom $i$ as follows:

$$m_{\mathcal{N}(i)}^{rad} = \sum_{j \neq i}^{N} exp(-\eta * (r_{ij} - \mu)^2) * f_c(r_{ij}), \tag{2}$$

where $N = |\mathcal{N}(i)|$ is the number of neighboring atoms, $r_{ij}$ is the distance between atoms $i$ and $j$, $\eta$ and $\mu$ are parameters controlling the width and center of the Gaussian. The cutoff function $f_c(\cdot)$ ensures that only significant neighbors are considered.

Similarly, the angular ACSF captures the angular relationships between atoms:

$$m_{\mathcal{N}(i)}^{ang} = 2^{1-\zeta} \sum_{j \neq i}^{N} \sum_{k \neq i, j}^{N} (1 + \lambda cos(\theta_{ijk}))^{\zeta} e^{-\eta * (r_{ij} - \mu)^2} e^{-\eta * (r_{ik} - \mu)^2} e^{-\eta * (r_{jk} - \mu)^2}$$
$$* f_c(r_{ij}) * f_c(r_{ik}) * f_c(r_{jk}), \tag{3}$$

where $\theta_{ijk}$ is the angle formed by atoms $i$, $j$, and $k$, and $\lambda$ adjusts the peak of the angular term. The parameter $\zeta$ controls the width of the angular distribution.

## 3 GEOMETRICALLY EQUIVARIANT GNNs

### 3.1 INVARIANT GEOMETRIC GNNs

Recent advancements in geometric graph models have introduced novel methods to achieve equivariance, with many focusing on scalarization rather than relying solely on group representation theory. In scalarization, geometric vectors are first transformed into invariant scalar quantities, which are then passed through multiple layers of multilayer perceptrons (MLPs). These scalars control the magnitude of the original vectors, which are subsequently reintroduced in their original directions to ensure equivariance. This method was pioneered in models like SchNet (Schütt et al., 2018) and DimeNet (Gasteiger et al., 2020b), though they applied it in a purely invariant manner. Building on these models, SphereNet (Liu et al., 2022) extended scalarized message passing by incorporating angular and torsional information, enabling the model to differentiate molecular chirality while maintaining overall invariance. Similarly, Radial Field (Köhler et al., 2020) introduced an equivariant version of this scalarization approach, although it focused exclusively on geometric vectors without considering node features.

Despite the relative simplicity of scalarization techniques, their effectiveness has been theoretically supported. Villar et al. (Villar et al., 2021) demonstrated that methods based on inner products could universally achieve equivariance. Expanding upon this foundation, GemNet (Klicpera et al., 2021) integrated richer geometric information, such as dihedral angles, into the message-passing framework, advancing the principles initially laid out by DimeNet. The key idea remains: multiplying an invariant scalar by an equivariant vector still results in an equivariant vector.

Several additional approaches to equivariant message passing have emerged, following these scalarization principles. For example, PaiNN (Schütt et al., 2021) and the Equivariant Transformer (Thölke & De Fabritiis, 2022a) enhance the invariant SchNet by using radial basis functions to project interatomic distances. These methods iteratively update both vector and scalar features during the message-passing process. GVPGNN (Jing et al., 2020), on the other hand, offers a stronger theoretical framework for achieving universal equivariance in message passing.

When constructing invariant geometric GNNs, there are two principal approaches for incorporating geometric invariance: distance-based invariant GNNs and many-body-based invariant GNNs. These approaches differ in how they encode geometric information, offering distinct strategies to capture molecular structures and interactions.

**Distance-based invariant GNNs:** Invariant geometric GNN layers aggregate scalar values from local neighborhoods by transforming geometric information into scalar quantities (Schütt et al., 2018). These scalar features are then updated from layer $t$ to $t + 1$ using trainable AGGREGATE and COMBINE functions, which operate as follows:

$$
\mathbf{m}_{ij}^{(t)} = \sum_{j \in \mathcal{N}(\rangle)} f_{\theta_1}(\mathbf{h}_j^{(t)}, ||\vec{\mathbf{x}}_{ij}||)
$$
$$
\mathbf{h}_i^{(t+1)} = \text{COMBINE}(\mathbf{h}_i^{(t)}, \mathbf{m}_{ij}^{(t)}),
$$
(4)

where $\mathbf{h}_i \in \mathbb{R}^z$ denote the scalar feature vector of node $i$, and $\vec{\mathbf{x}}_{ij} \in \mathbb{R}^d$ represents the relative positional vector: $||\vec{\mathbf{x}}_{ij}|| = ||\vec{\mathbf{x}}_i - \vec{\mathbf{x}}_j||$, where $\vec{\mathbf{x}}_i \in \mathbb{R}^d$ denotes the 3D coordinate vector of node $i$.

**Distances and many-body-based invariant GNNs:** To overcome the limitations in geometric expressiveness posed by distance-based message passing, recent invariant GNN models (Shuaibi et al., 2021; Wang & Zhang, 2022; Wang et al., 2022) emphasize the inclusion of scalar quantities derived from higher-order interactions, extending beyond pairwise connections (such as triplets of atoms).

$$
\mathbf{m}_{ij}^{(t)} = \sum_{j \in \mathcal{N}(\rangle)} f_{\theta_1}\Big(\mathbf{h}_i^{(t)}, \mathbf{h}_j^{(t)}, d_{ij}, \sum_{k \in \mathcal{N}(j) \setminus \{i\}} f_{\theta_2}\Big(\mathbf{h}_j^{(t)}, \mathbf{h}_k^{(t)}, d_{ij}, \angle ijk\Big)\Big)
$$
$$
\mathbf{h}_i^{(t+1)} = \text{COMBINE}(\mathbf{h}_i^{(t)}, \mathbf{m}_{ij}^{(t)}),
$$
(5)

where $d_{ij} = ||\vec{\mathbf{x}}_{ij}||$ represents the distance between atoms $i$ and $j$, and bond angles $\angle ijk = \angle(\vec{\mathbf{x}}_{ij}, \vec{\mathbf{x}}_{ik})$ are defined for atom triplets (i.e., 3-body order interactions).

## 3.2 IRREDUCIBLE REPRESENTATIONS

Equivariant neural networks are designed to process geometric tensors, such as type-L vectors, to maintain equivariance (Thomas et al., 2018; Miller et al., 2020; Batzner et al., 2022; Brandstetter et al., 2022; Thölke & De Fabritiis, 2022b). A key technique employed is the use of geometric functions derived from spherical harmonics and irreducible representations (irreps), that networks behave equivariantly in 3D Euclidean space.

Different approaches within this framework vary in how they implement equivariant operations, particularly in their message-passing mechanisms. Both TFN (Thomas et al., 2018) and NequIP (Batzner et al., 2022) use linear message-passing strategies, where equivariant message updates are achieved through convolutional layers. NequIP further enhances the equivariant architecture by introducing gating mechanisms that modulate the messages in a structured manner, improving model performance. In contrast, SEGNN (Brandstetter et al., 2022) deviates by employing non-linear message-passing mechanisms on irreducible representations, incorporating a similar gating mechanism, but extending beyond linear messages to achieve more expressive power. Another prominent architecture, the SE(3)-Transformer (Fuchs et al., 2020), builds on the dot-product attention mechanism, applying it to type-L vectors in an equivariant manner. Subsequent models (Thölke & De Fabritiis, 2022b; Le et al., 2022a) refined these Transformers, tailoring them specifically to work with type-0 and type-1 vectors while maintaining the attention-based framework.

Central to these architectures is the ability to update higher-order spherical tensor features in an equivariant manner (Thomas et al., 2018; Batatia et al., 2022; Fuchs et al., 2020; Brandstetter et al., 2021). For instance, TFN layers(Thomas et al., 2018) use higher order spherical tensors as node features $\tilde{\mathbf{h}}_i \in \mathbb{R}^{2l+1 \times z}$, where $l$ represents the order of the tensor $l = 0, \ldots l = L$. The first and second orders represent the scalar features $\mathbf{h}_i \in \mathbb{R}^z$ and vector features $\vec{\mathbf{f}}_i \in \mathbb{R}^z$, respectively. The higher order tensors $\tilde{\mathbf{h}}_i$ are updated through Clebsch-Gordan tensor products (Coope, 1970) $\otimes_{cg}^w$ of local neighborhood features $\tilde{\mathbf{h}}_j$ for all $j \in N(i)$ with higher order spherical harmonic expansion of displacement $\mathbf{Y}^{(l)}(\tilde{\mathbf{x}}_{ij}) \in \mathbb{R}^{2l+1}$, where $\mathbf{Y}^{(l)}$ represents the higher order spherical harmonic

representations and $\tilde{\mathbf{x}}_{ij} = \frac{\vec{\mathbf{x}}_{ij}}{||\vec{\mathbf{x}}_{ij}||}$ represents the relative displacement. The message passing scheme is defined as below:

$$\mathbf{m}_{ij}^{(t)} = \sum_{j \in N(i)} \mathbf{Y}^{(l)}(\tilde{\mathbf{x}}_{ij}) \otimes_{cg}^{w} \tilde{\mathbf{h}}_{j}^{(t)}$$

$$\tilde{\mathbf{h}}_{i}^{(t+1)} = \text{COMBINE}(\tilde{\mathbf{h}}_{i}^{(t)}, \mathbf{m}_{ij}^{(t)}),$$

(6)

where $w = \psi_{rbf}(\mathbf{h}_j, ||\vec{\mathbf{x}}_{ij}||)$ is the tensor product weight and $\psi_{rbf}$ is the learned radial basis function.

This combination of irreducible representations, spherical harmonics, and tensor products provides a robust framework for building equivariant GNNs capable of maintaining geometric consistency across varying transformations, offering expressive models that perform well on tasks requiring 3D rotational and translational invariance.

## 3.3 SELF-ATTENTION TRANFORMERS

In a self-attention module (Vaswani, 2017), each node with associated features is mapped into a query and a set of key-value pairs to produce an output. The output is calculated as a weighted sum of the values, where the weights are based on the similarity between the query and the corresponding key. These modules are relatively easy to implement and offer significant design flexibility (Baek et al., 2021; Jumper et al., 2021), which has led to their broad application across various domains, including language modeling (Vaswani, 2017; Devlin, 2018) and graph-based tasks (Dwivedi & Bresson, 2020; Veličković et al., 2018).

Recently, SE(3)- and E(3)-equivariant self-attention modules have been introduced. In (Fuchs et al., 2020), constructs SE(3)-equivariant query, key, and value representations using irreducible representations. In E(3)-equivariant models, the L2-norm of the 3D coordinate vector differences between nodes is frequently used as an invariant feature. Some methods generate queries, keys, and values from scalar features and combine the query-key similarity with invariant features through addition (Maziarka et al., 2020) or multiplication (Morehead et al., 2022; Le et al., 2022b). These operations enhance the model's ability to handle geometric transformations effectively.

Additionally, some models adopt a more generalized self-attention approach by calculating attention weights directly from both scalar features and invariant properties. This technique, as seen in works such as (Satorras et al., 2021; Köhler et al., 2020; Schütt et al., 2017), provides greater flexibility in capturing complex interactions between nodes, leading to improved performance in graph-based tasks. By leveraging both scalar and invariant features in the attention mechanism, these models can more effectively encode the geometric and topological structure of the data, resulting in more powerful representations.

## 4 EQUIVARIANT GRAPH SELF-ATTENTION TRANSFORMER (EG-SAT)

We introduce a new class of Atom-Centered Symmetry Functions (ACSFs) (Schutt et al., 2018), referred to as Attention-based Atom-Centered Symmetry Functions (AACSFs), which maintain roto-translational invariance. In addition, we propose a novel Equivariant Graph Self-Attention Transformer (EG-SAT). We provide a theoretical analysis of how EG-SAT can be designed to achieve equivariance, ensuring that it respects the necessary symmetry properties for accurate modeling of 3D molecular structures. This design enables EG-SAT to effectively capture higher-order interactions and complex spatial relationships among atoms while preserving the model's ability to generalize across different molecular environments.

### 4.1 ATTENTION-BASED ATOM-CENTERED SYMMETRY FUNCTIONS

The primary limitations of Atom-Centered Symmetry Functions (ACSFs) become evident when dealing with molecules that exhibit diverse chemical elements. First, ACSFs do not scale efficiently with element diversity; they require an increasing number of radial and angular functions to represent all possible element pairs and triples. As the number of elements in a system increases, the number of necessary symmetry functions grows exponentially, complicating the descriptor vector size (Gastegger et al., 2018). This leads to a significant rise in computational overhead for both model training and

the transformation of Cartesian coordinates. Consequently, the computational demands become particularly burdensome for applications such as high-throughput screening and molecular dynamics simulations. As a result, ACSFs are less practical for systems with complex chemical compositions due to their poor scalability and high computational costs.

To address these shortcomings of traditional ACSFs, we propose a novel modification to this type of descriptor. Instead of employing distinct functions for each combination of elements, we incorporate the chemical composition of the environment in a more implicit manner. This is achieved by integrating element-dependent attention functions into the radial and angular distribution equations (Eq: 2 and 3), which allows for a more efficient representation. Specifically, the radial attention-based ACSFs are defined as:

$$\mathbf{m}_{\mathcal{N}(i),rad}^{(t)} = \sum_{j \neq i}^{N} g_\phi(\mathbf{h}_i^{(t)}, \mathbf{h}_j^{(t)}) * a_\theta(\mathbf{h}_i^{(t)}, d_{ij}, e_{ij}) * \sigma(exp(-\eta * (d_{ij} - \mu)^2)) * f_c(\mathbf{h}_j^{(t)}), \quad (7)$$

where $g_\phi(\mathbf{h}_i^{(t)}, \mathbf{h}_j^{(t)})$ = Sigmoid(GRU$_{\tilde{\theta}}$(Concat($\mathbf{h}_i^{(t)}, \mathbf{h}_j^{(t)}$))) is a gating function and $a_\theta(\mathbf{h}_i^{(t)}, d_{ij}, e_{ij})$ = Softmax($Q * K^T$) is an attention function. Here, $Q = \mathbf{h}_i^{(t)} W_Q$ and $K =$ Concat($e_{ij}, d_{ij}) W_k$, and $\sigma$ is the Softplus activation function. The pairwise distance $d_{ij} = ||\vec{\mathbf{x}}_i - \vec{\mathbf{x}}_j||^2$, where $\vec{\mathbf{x}}_i$ and $\vec{\mathbf{x}}_j$ are the 3D coordinates of atom $i$ and $j$, respectively.

The radial attention-based ACSF function offers a more expressive and flexible approach to modeling the radial relationships between atoms compared to traditional methods. By integrating attention mechanisms, this function dynamically weights the contributions of different atom pairs based on their distances and atomic features. Additionally, the gating mechanism enhances flexibility by modulating the influence of each atom pair, while the Gaussian radial decay ensures that interactions are focused on relevant neighboring atoms based on spatial proximity.

Building on this foundation, we further define the angular attention-based ACSFs as:

$$\mathbf{m}_{\mathcal{N}(i),ang}^{(t)} = \gamma * \sum_{j \neq i}^{N} \sum_{k \neq i,j}^{N} g_\phi(\mathbf{h}_i^{(t)}, \mathbf{h}_j^{(t)}, \mathbf{h}_k^{(t)}) * a_\theta(\mathbf{h}_i^{(t)}, d_{ij}, d_{ik}, d_{jk}, \beta_{ijk}) * (1 + \lambda cos(\beta_{ijk}))^\zeta *$$
$$\sigma\left(e^{-\eta * (d_{ij} - \mu)^2} * f_c(\mathbf{h}_i^{(t)}, \mathbf{h}_j^{(t)})\right) * \sigma\left(e^{-\eta * (d_{ik} - \mu)^2} * f_c(\mathbf{h}_i^{(t)}, \mathbf{h}_k^{(t)})\right) *$$
$$\sigma\left(e^{-\eta * (d_{jk} - \mu)^2} * f_c(\mathbf{h}_j^{(t)}, \mathbf{h}_k^{(t)})\right),$$
$$(8)$$

where $g_\phi(\mathbf{h}_i^{(t)}, \mathbf{h}_j^{(t)}, \mathbf{h}_k^{(t)})$ = Sigmoid(GRU$_{\tilde{\theta}}$(Concat($\mathbf{h}_i^{(t)}, \mathbf{h}_j^{(t)}, \mathbf{h}_k^{(t)}$))) is a gating function and $a_\theta(\mathbf{h}_i, d_{ij}, d_{ik}, d_{jk}, cos(\beta_{ijk}))$ = Softmax($Q * K^T$) is an attention function. Here, $Q = \mathbf{h}_i^{(t)} W_Q$ and $K =$ Concat($d_{ij}, d_{ik}, d_{jk}, cos(\beta_{ijk})) W_k$, and $\sigma$ is the Softplus activation function. The $\gamma, \lambda, \zeta, \eta$ and $\mu$ control various aspects of the model's behavior: $\gamma$ scales the overall contribution of the attention mechanism, $\lambda$ adjusts the weight of the cosine angular term, $\zeta$ modulates the influence of angular features, $\eta$ controls the width of the radial basis function, and $\mu$ shifts the center of the distance-based Gaussian. We use $f_c(\mathbf{h}_i^{(t)}, \mathbf{h}_j^{(t)})$ to represent the feature difference between neighboring atoms, where $f_c(\mathbf{h}_i^{(t)}, \mathbf{h}_j^{(t)}) = \mathbf{h}_i^{(t)} - \mathbf{h}_j^{(t)}$. This difference measures how the features of neighboring atoms change as the message-passing process progresses through the layers.

The key novelty of angular attention-based ACSFs approach lies in the combination of attention and gating mechanisms within the angular-based ACSFs. The attention mechanism dynamically weights the relative importance of angular features, while the gating mechanism regulates the flow of information across different atom triplets. This dual integration makes our model highly flexible and capable of capturing nuanced geometric relationships between atoms-an advancement that traditional ACSF methods are unable to achieve.

Finally, we compute the final radial attention-based outcome, $\mathbf{h}_{i,rad}^{(t+1)} = \text{MLP}((1+\epsilon)\mathbf{h}_i^{(t)} + \mathbf{m}_{\mathcal{N}(i),rad}^{(t)})$. Similarly, we can compute the angular attention-based outcome, $\mathbf{h}_{i,ang}^{(t+1)} = \text{MLP}((1+\epsilon)\mathbf{h}_i^{(t)} + \mathbf{m}_{\mathcal{N}(i),ang}^{(t)})$. Then, we concatenate these two outcomes, $\mathbf{h}_i^{(t+1)} = \text{Concat}(\mathbf{h}_{i,rad}^{(t+1)}, \mathbf{h}_{i,ang}^{(t+1)})$ to compute the final representation.

| Property | $\alpha$ | $\Delta\epsilon$ | $\epsilon_{HOMO}$ | $\epsilon_{LUMO}$ | $\mu$ | $C_v$ | G | H | $R^2$ | U | $U_0$ | ZPVE |
|---|---|---|---|---|---|---|---|---|---|---|---|---|
| SchNet* | 0.235 | 63 | 41 | 34 | 0.033 | 0.033 | 14 | 14 | 0.073 | 19 | 14 | 1.70 |
| Cormorant | 0.085 | 61 | 34 | 38 | 0.038 | 0.026 | 20 | 21 | 0.961 | 21 | 22 | 2.03 |
| DimeNet* | 0.047 | 35 | 28 | 20 | 0.029 | 0.025 | 9 | 8 | 0.331 | 8 | 8 | 1.29 |
| NMP* | 0.092 | 69 | 43 | 38 | 0.030 | 0.040 | 19 | 17 | 0.180 | 20 | 20 | 1.50 |
| TFN | 0.223 | 58 | 40 | 38 | 0.064 | 0.101 | - | - | - | - | - | - |
| LieConv | 0.084 | 49 | 30 | 25 | 0.032 | 0.038 | 22 | 24 | 0.800 | 19 | 19 | 2.28 |
| L1Net* | 0.088 | 68 | 46 | 35 | 0.043 | 0.031 | 14 | 14 | 0.354 | 14 | 13 | 1.56 |
| SEGNN | 0.060 | 42 | 24 | 21 | 0.023 | 0.031 | 15 | 16 | 0.660 | 13 | 15 | 1.62 |
| EGNN | 0.071 | 48 | 29 | 25 | 0.029 | 0.031 | 12 | 12 | 0.106 | 12 | 11 | 1.55 |
| SE(3)-Transformer | 0.142 | 53 | 35 | 33 | 0.051 | 0.054 | - | - | - | - | - | - |
| **EG-SAT** | **0.042** | **33** | **22** | **21** | **0.020** | **0.023** | 15 | 16 | 0.280 | **6** | **6** | **1.50** |

Table 1: Mean absolute error (MAE) for 12 quantum chemical properties on the QM9 dataset. Models marked with an asterisk (*) use a different training/validation/testing data split. All results are sourced from their respective original papers.

**Complexity Analysis:** EG-SAT is efficient in terms of computation, with both its time and memory requirements growing linearly with the number of edges in the graph. The time complexity is $O(t(anzm + aem))$ and the space complexity is $O(e)$, where $a$ indicates the number of attention heads, $e$ is the number of edges, $n$ is the number of nodes, $t$ is the number of layers, and $z$ and $m$ represent the dimensions of the input and output feature vectors, respectively.

**Theorem 1.** *Radial AACSFs, Angular AACSFs, and EG-SAT are all SE(3)-equivariant.*

## 5 NUMERICAL EXPERIMENTS

In this section, we evaluate the performance of our EG-SAT transformer on the QM9 (Gilmer et al., 2017; Wu et al., 2018) and MD17 (Chmiela et al., 2017) datasets for quantum molecular property prediction tasks.

### 5.1 QM9

The QM9 dataset consists of more than 130,000 small organic molecules with quantum chemical properties calculated using Density Functional Theory (DFT). The molecules contain up to nine heavy atoms, including elements such as hydrogen (H), carbon (C), nitrogen (N), oxygen (O), and fluorine (F). This dataset is widely used for training and evaluating machine learning models in molecular property prediction tasks. In our study, we utilized a more complex data partitioning scheme, where 100,000 molecules were allocated for training, 18,000 for validation, and 13,000 for testing, following the setup used by Cormorant (Anderson et al., 2019). During training, the model optimizes the mean absolute error (MAE) between the predicted outputs and the true molecular property labels.

**Baselines:** We compare our method against ten baseline approaches: SchNet (Schütt et al., 2017), Cormorant (Anderson et al., 2019), DimeNet Gasteiger et al. (2020a), NMP (Gilmer et al., 2017), TFN (Thomas et al., 2018), LieConv (Finzi et al., 2020), L1Net (Miller et al., 2020), SEGNN (Brandstetter et al., 2021), EGNN (Satorras et al., 2021), and SE(3)-Transformer(Fuchs et al., 2020).

**Experimental setup for QM9:** We employ the Adam optimizer (Kingma, 2014). Our model is trained for 500 epochs using a learning rate of 0.001, batchsize 64, $\alpha = 1$, $\mu = 0.01$, $\zeta = 2$, $\eta = 1$, and $\lambda = 0.5$. We select the optimal weight decay from $\{0.0001, 0.0002, \ldots, 0.0009\}$ and hidden units from $\{64, 128, 256, 512\}$. For dropout rate, the best values for each property on QM9 are chosen from $\{0.1, 0.2, ..., 0.6\}$.

### 5.2 MD17

The MD17 dataset contains molecular dynamics simulations of small organic molecules, including atomic trajectories computed using DFT. We utilize MD17 to assess the performance of EG-SAT in predicting molecular forces, a key requirement for molecular dynamics tasks. With a training set comprising 1,000 samples, the remaining data is set aside for testing the model's accuracy in predicting energies and forces. The model outputs scalar energy (E), and the forces are derived

| | Molecule | Aspirin | Ethanol | Malonaldehyde | Naphthalene | Salicylic acid | Toluene | Uracil |
|---|---|---|---|---|---|---|---|---|
| SchNet | energy | 0.37 | 0.08 | 0.13 | 0.16 | 0.20 | 0.12 | 0.14 |
| | forces | 1.35 | 0.39 | 0.66 | 0.58 | 0.85 | 0.57 | 0.56 |
| DimeNet | energy | 0.204 | 0.064 | 0.104 | 0.122 | 0.134 | 0.102 | 0.115 |
| | forces | 0.499 | 0.230 | 0.383 | 0.215 | 0.374 | 0.216 | 0.301 |
| PaiNN | energy | 0.159 | 0.063 | 0.091 | 0.117 | 0.114 | 0.098 | 0.104 |
| | forces | 0.371 | 0.230 | 0.319 | 0.151 | 0.221 | 0.203 | 0.105 |
| NequIP | energy | - | - | - | - | - | - | - |
| | forces | 0.348 | 0.208 | 0.337 | 0.096 | 0.238 | 0.101 | 0.172 |
| PhysNet | energy | 0.230 | 0.059 | 0.094 | 0.142 | 0.126 | 0.100 | 0.108 |
| | forces | 0.605 | 0.160 | 0.219 | 0.310 | 0.337 | 0.191 | 0.218 |
| sGDML | energy | 0.19 | 0.07 | 0.10 | 0.12 | 0.12 | 0.10 | 0.11 |
| | forces | 0.68 | 0.33 | 0.41 | 0.11 | 0.28 | 0.14 | 0.24 |
| NewtonNet | energy | 0.168 | 0.078 | 0.096 | 0.118 | 0.115 | 0.094 | 0.107 |
| | forces | 0.348 | 0.264 | 0.323 | 0.084 | 0.197 | 0.088 | 0.149 |
| FCHL19 | energy | 0.182 | 0.054 | 0.081 | 0.117 | 0.114 | 0.098 | 0.104 |
| | forces | 0.478 | 0.136 | 0.245 | 0.151 | 0.221 | 0.203 | 0.105 |
| EG-SAT | **energy** | **0.150** | **0.040** | **0.080** | **0.100** | 0.130 | 0.100 | **0.100** |
| | **forces** | **0.200** | **0.120** | **0.200** | **0.080** | 0.220 | 0.200 | **0.101** |

Table 2: Mean absolute error (MAE) metrics for energies (kcal/mol) and forces (kcal/mol/Å) on the MD17 dataset. All results are derived from the original publications.

by differentiating this energy with respect to atomic positions, $F_i = -\partial E/\partial r_i$. During training, the model simultaneously minimizes the energy and force loss, weighted by factors of 1 and 100, respectively.

**Baselines:** We compare our method against eight baseline approaches: SchNet (Schütt et al., 2017), DimeNet (Gasteiger et al., 2020a), PaiNN (Schütt et al., 2021), NequIP (Batzner et al., 2022), PhysNet (Unke & Meuwly, 2019), sGDML (Chmiela et al., 2019), NewtonNet (Haghighatlari et al., 2022), and FCHL19 (Christensen et al., 2020).

**Experimental setup for MD17:** We use the Adam optimizer (Kingma, 2014). Our model is trained for 200 epochs using a learning rate of 0.0001, batchsize 1, $\alpha = 1$, $\mu = 0.001$, $\zeta = 2$, $\eta = 1$, and $\lambda = 0.8$. We select the optimal weight decay from $\{0.0001, 0.0002, \ldots, 0.0009\}$ and hidden units from $\{64, 128, 256, 512\}$. For dropout rate, the best values for each dataset on MD17 are chosen from $\{0.1, 0.2, ..., 0.6\}$.

## 6 CONCLUSION

In conclusion, our work introduces a novel framework for modeling 3D molecular structures with improved scalability, accuracy, and geometric representation. By developing Attention-based Atom-Centered Symmetry Functions (AACSFs) and the Equivariant Graph Self-Attention Transformer (EG-SAT), we provide a robust solution to the limitations of traditional methods. The integration of element-specific attention mechanisms and gating functions allows EG-SAT to capture higher-order atomic interactions and intricate spatial relationships more effectively than existing models. Our approach addresses the scalability issues associated with ACSFs, offering an efficient way to handle diverse chemical elements while preserving the necessary symmetry properties for molecular modeling. Validation on the QM9 and MD17 datasets demonstrates the better performance of EG-SAT, making it a valuable tool for applications in quantum chemistry and drug discovery. This work paves the way for more accurate and scalable models in the study of molecular interactions and quantum mechanical properties.

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

# A APPENDIX

**Theorem 1.** *Radial AACSFs, Angular AACSFs, and EG-SAT are all SE(3)-equivariant.*

*Proof.* Let $\vec{\mathbf{z}} \in \mathbb{R}^3$ be a translation vector, $\mathbf{Q} \in \mathbb{R}^{3 \times 3}$ be an orthogonal matrix. We will consider the feature vector $\mathbf{h}_i^{(t)}$, which is already invariant under the Euclidean group SE(3).

**Translation Equivariance:** Assume the input 3D coordinates are transformed by a translation vector $\vec{\mathbf{z}}$: $\vec{\mathbf{x}}_i' = \vec{\mathbf{x}}_i + \vec{\mathbf{z}}$ and $\vec{\mathbf{x}}_j' = \vec{\mathbf{x}}_j + \vec{\mathbf{z}}$. Then, the pairwise distance transforms as follows: $d_{ij}' = ||\vec{\mathbf{x}}_i' - \vec{\mathbf{x}}_j'||^2 = ||(\vec{\mathbf{x}}_i + \vec{\mathbf{z}}) - (\vec{\mathbf{x}}_j + \vec{\mathbf{z}})||^2 = ||\vec{\mathbf{x}}_i - \vec{\mathbf{x}}_j||^2 = d_{ij}$. Thus, $d_{ij}$ remains unchanged under translation, leading to:

$$\mathbf{m}_{\mathcal{N}(i),rad}^{(t)'} = \sum_{j \neq i}^{N} g_\phi(\mathbf{h}_i^{(t)}, \mathbf{h}_j^{(t)}) * a_\theta(\mathbf{h}_i^{(t)}, d_{ij}, e_{ij}) * \sigma(exp(-\eta * (d_{ij} - \mu)^2)) * f_c(\mathbf{h}_j^{(t)}) = \mathbf{m}_{\mathcal{N}(i),rad}^{(t)}$$

**Rotation Equivariance:** Now, assume a rotation is applied using an orthogonal matrix $\mathbf{Q}$: $\vec{\mathbf{x}}_i' = \mathbf{Q}\vec{\mathbf{x}}_i$ and $\vec{\mathbf{x}}_j' = \mathbf{Q}\vec{\mathbf{x}}_j$. Then, the distances transform as: $d_{ij}' = ||\mathbf{Q}\vec{\mathbf{x}}_i - \mathbf{Q}\vec{\mathbf{x}}_j||^2 = ||\mathbf{Q}(\vec{\mathbf{x}}_i - \vec{\mathbf{x}}_j)||^2 = ||\vec{\mathbf{x}}_i - \vec{\mathbf{x}}_j||^2 = d_{ij}$. The outputs $g_\phi$ and $a_\theta$ are also invariant under such transformations, as their formulations do not depend on the absolute positions of the nodes. Hence, we have shown that:

$$\mathbf{m}_{\mathcal{N}(i),rad}^{(t)'} = \mathbf{m}_{\mathcal{N}(i),rad}^{(t)}$$

Thus, Radial AACSFs is SE(3)-equivariant.

Similarly, we can prove the SE(3)-equivariance for the angular AACSFs. Under the same translation $\mathbf{z}$: $d_{ij}' = d_{ij}, d_{ik}' = d_{ik}, d_{jk}' = d_{jk}$. The angle $\beta_{ijk}$ between vectors remains unchanged since $cos(\beta_{ijk}') = cos(\beta_{ijk})$. Thus, we can show $\mathbf{m}_{\mathcal{N}(i),ang}^{(t)'} = \mathbf{m}_{\mathcal{N}(i),ang}^{(t)}$. Applying the same rotation $\mathbf{Q}$: $\vec{\mathbf{x}}_i' = \mathbf{Q}\vec{\mathbf{x}}_i, \vec{\mathbf{x}}_k' = \mathbf{Q}\vec{\mathbf{x}}_k, \vec{\mathbf{x}}_j' = \mathbf{Q}\vec{\mathbf{x}}_j$. The distances transform as previously shown, and the angular relationships are preserved under rotation. Hence, the equations remain invariant.

The output of EG-SAT depends on the radial and angular attension-based ACSFs: $\mathbf{h}_i^{(t+1)} = $ Concat$(\mathbf{h}_{i,rad}^{(t+1)}, \mathbf{h}_{i,ang}^{(t+1)})$. Thus, we can conclude that EG-SAT is also E(n)-equivariant. Therefore, both the radial and angular ACSFs are equivariant under SE(3)-transformations, concluding the proof. □

