# OpenReview forum: "Equivariant Graph Self-Attention Transformer for Learning Higher-Order Interactions in 3D Molecular Structures"
_ICLR.cc/2025/Conference — ICLR 2025 Conference Withdrawn Submission_

### Official Review · Reviewer_JAQn · 2024-10-24

**Soundness:** 1
**Presentation:** 2
**Contribution:** 1
**Rating:** 3
**Confidence:** 5

**Summary:**

Previous geometric graph neural networks generally exhibit poor scalability when dealing with large molecular structure data. To address this issue, this paper proposes improvements to the traditional atom-centered symmetry functions by incorporating self-attention mechanisms to integrate both angular and radial information. This approach enhances the scalability of the graph neural network while preserving rotational and translational invariance.

**Strengths:**

The paper introduces the related work and background knowledge of equivariant graph neural networks in a detailed and clear manner, allowing readers to quickly establish relevant domain knowledge.

**Weaknesses:**

1. The writing and formatting of the paper are somewhat rough. The size of Figure 1 and the font of Figure 2 both require adjustments. The caption of the table should be placed above the table. Additionally, Section 4 contains only a portion of the content, yet it is labeled with a subsection title '4.1,' which seems redundant. The first sentence of the abstract uses 'their' without a clear referent, among other issues. These problems make the article difficult to read and do not meet the standard of a top conference paper.
2. This paper resembles a review of equivariant graph neural networks, and its actual contributions are not aligned with what is claimed in the introduction. The article devotes a significant amount of space to background knowledge and related work, only presenting the proposed method towards the end of page 7. Given the structure, it would be more appropriate to submit this as a review paper.
3. The experimental performance of the proposed method is not promising, and the baseline methods used for comparison are somewhat outdated, mostly from before 2021. Several well-known methods in the field, such as Equiformer[1], are missing from the comparison. As a result, the experiments do not effectively demonstrate the validity of the proposed method.
4. There seems to be an error in Equation 5, where $d_{ij}$ should be $d_{jk}$.
5. The innovation of the proposed method is rather limited. Introducing the attention mechanism into graph neural networks is not particularly novel, and I am curious about the differences and connections between the proposed method and existing approaches like GAT[2].

[1] Equiformer: Equivariant graph attention transformer for 3d atomistic graphs.

[2] Graph attention networks.

**Questions:**

Please refer to Weaknesses 5.

---

### Official Review · Reviewer_1Hn3 · 2024-10-24

**Soundness:** 2
**Presentation:** 2
**Contribution:** 1
**Rating:** 3
**Confidence:** 4

**Summary:**

This manuscript describes a novel machine learning model called the Equivariant Graph Self-Attention Transformer (EG-SAT), which is designed to overcome challenges in capturing geometric and relational structures of molecules. By using Atom-Centered Symmetry Functions (ACSFs), EG-SAT captures molecular geometry through higher-order representations, modeling both the radial and angular distributions of neighboring atoms within a certain cutoff distance. This allows the model to gain a nuanced understanding of molecular interactions, which benefits the geometric information preservation.

In the experiment, the model is validated on the QM9 and MD17 datasets. Compared with multiple methods, EG-SAT shows state-of-the-art performance, especially in predicting quantum mechanical properties.

**Strengths:**

This paper is well-organized with clear writting.

**Weaknesses:**

Lack of Novelty:
1. Equivariance Claim: The proposed method, Equivariant Graph Self-Attention Transformer (EG-SAT), is supposed to be equivariant, which the authors aim to achieve by encoding the geometric information of 3D graphs in Euclidean space. However, this is a commonly used approach seen in models such as SchNet, DimeNet, and SphereNet, which are not only equivariant but also invariant. Therefore, the claim of "equivariance" is insufficient as a primary contribution of this method. Given that the model is invariant, introducing the concept of irreducible representations (irreps) in Section 3 is unnecessary. The irreps of SO(3) are typically not used in invariant graph neural networks (GNNs).

2. Graph Transformer: The model is based on the graph transformer architecture, which is also widely used in the field. This, again, is not sufficient to be considered a significant contribution of the work.

3. Scalability of ACSFs: The authors claim to address scalability issues in Atom-Centered Symmetry Functions (ACSFs) using attention-based mechanisms through GRU blocks to approximate interactions between atoms. However, attention mechanisms for interaction have already been considered within the graph transformer framework. The authors should further explain the necessity of introducing this specific mechanism in their approach.

Insufficient Experiments:

1.Outdated Comparisons: The experiments compare the proposed method with multiple other approaches. However, many of these comparison methods are outdated. The authors should benchmark their model against more recent methods such as SphereNet and Molformer. Moreover, since the method is theoretically invariant, there is little need to compare it with numerous equivariant methods.

**Questions:**

Suggestions:
1. Improve the novelty of the method.
2. Make more comparison with newer invariant methods to demonstrate the effectiveness and robustness.
3. Adjust the framework of the manuscript, reduce the length of background and related work, and strengthen the correlation with the proposed method.

---

### Official Review · Reviewer_TByL · 2024-11-02

**Soundness:** 2
**Presentation:** 2
**Contribution:** 2
**Rating:** 3
**Confidence:** 4

**Summary:**

This paper aims to improve computational efficiency while preserving equivariance under Euclidean transformations for 3D molecules of varying sizes. By introducing the ACSFs, the authors propose the Equivariant Graph Self-Attention Transformer (EG-SAT), which leverages both geometric and relational features while maintaining roto-translational invariance. The theoretical analysis and time complexity are presented. Experimental results on the QM9 and MD17 datasets demonstrate the effectiveness of the proposed method.

**Strengths:**

- The paper addresses a significant research question and proposes a method to solve it.
- The paper employs a self-attention Transformer to achieve equivariance while reducing computational costs.
- Comprehensive background information is provided, making the paper accessible and easy to follow.
- Experiments on the QM9 and MD17 datasets demonstrate the effectiveness of the proposed method.

**Weaknesses:**

- The specific research problem addressed by the paper is unclear. While it provides comprehensive background information on molecular learning, symmetry, invariance, irreps, and ACSFs, the research problem is not clearly defined. In Sec. 4.1, the authors highlight the limitations of ACSFs but do so without detailed discussion or analysis to clarify the issue.
- The paper contains excessive information that may obscure its primary focus. For instance, the authors introduce irreps, but there is limited mention of its relevance within the method or analysis.
- Compared to ACSFs, the proposed EG-SAT still faces challenges in computational complexity. While the paper provides a complexity analysis for the proposed method, it lacks a direct comparison with ACSFs. Equations 7 and 8 do not offer computational savings relative to Equations 4 and 5.
- Overall, the content does not sufficiently support the contribution claims in Sec. 1. The paper lacks novelty and significant contributions.

**Questions:**

- The work primarily addresses the limitations of ACSFs. However, why are ACSFs not included in the comparison?
- What is the computational complexity of ACSFs?
- Can visualizations or case studies be provided to demonstrate that the method achieves equivariance?
- What criteria are used to select baselines? SchNet and DimeNet are invariant networks, so why are they chosen? Why do the comparison methods differ for QM9 and MD17? More recently proposed methods may also warrant comparison.
- What is meant by the gating mechanism in the paper, and what improvement does it offer over ACSFs?
- What challenges are encountered when integrating the self-attention Transformer, and what improvements does this integration provide?
- What are the limitations of the current work, and what steps could improve efficiency?

---

### Official Review · Reviewer_7Sfw · 2024-11-04

**Soundness:** 2
**Presentation:** 3
**Contribution:** 2
**Rating:** 3
**Confidence:** 5

**Summary:**

The paper proposes EG-SAT (Equivariant Graph Self-Attention Transformer), a novel approach for learning 3D molecular structures. The key contribution is the introduction of Attention-based Atom-Centered Symmetry Functions (AACSFs) that integrate both radial and angular information while maintaining roto-translational invariance. The model improves upon traditional ACSFs by incorporating element-specific attention mechanisms and addresses scalability challenges through attention-based mechanisms. The authors validate their approach on QM9 and MD17 datasets, demonstrating competitive performance in predicting quantum mechanical properties and molecular forces.

**Strengths:**

1. The paper attempts to address the scalability limitations of traditional ACSFs through attention mechanisms, providing a potentially interesting direction for future research in this area.

2. The mathematical formulation of the model's equivariance properties is presented in a structured manner with supporting proofs in the appendix, making the theoretical aspects accessible.

3. The implementation details are documented clearly with hyperparameters and architectural specifications, which aids in potential reproduction of the results.

**Weaknesses:**

1. Incomplete baselines for all the datasets presented in the paper. For QM9 dataset, authors didn't include recent works such as Spherenet [1], Equiformer(V2) [2,3], LEFTNet [4], SaVeNet [5], and Geoformer [6] to name few. Although the authors cited Equiformer, they didn't compare the results on QM9.

2. The baselines are on all small molecular tasks on QM9 and MD17 datasets. Therefore, limiting the applicability of the proposed methods.

3. Complexity Analysis: While the authors claim linear complexity with respect to the number of edges, this seems inconsistent with the use of angular information ($\beta_{ijk}$) which typically involves triplet interactions. The current analysis doesn't adequately justify how the model maintains linear complexity despite considering all possible triplets.

4. Empirical Validation of Efficiency Claims: Despite emphasizing computational efficiency and suitability for high-throughput screening, the paper lacks empirical evidence comparing computational costs with baseline methods.


[1] Liu, Y. _et al._ (2022) ‘Spherical Message Passing for 3D Molecular Graphs’, in _International Conference on Learning Representations_. Available at: https://openreview.net/forum?id=givsRXsOt9r.

[2] Liao, Y.-L. and Smidt, T. (2022) ‘Equiformer: Equivariant Graph Attention Transformer for 3D Atomistic Graphs’. Available at: https://openreview.net/forum?id=_efamP7PSjg.

[3] Liao, Y.-L. et al. (2024) ‘EquiformerV2: Improved Equivariant Transformer for Scaling to Higher-Degree Representations’, in The Twelfth International Conference on Learning Representations. Available at: https://openreview.net/forum?id=mCOBKZmrzD.

[4] Du, W. et al. (2023) ‘A new perspective on building efficient and expressive 3D equivariant graph neural networks’, in Thirty-seventh Conference on Neural Information Processing Systems. Available at: https://openreview.net/forum?id=hWPNYWkYPN.

[5] Aykent, S. and Xia, T. (2023) ‘SaVeNet: A Scalable Vector Network for Enhanced Molecular Representation Learning’, in Thirty-seventh Conference on Neural Information Processing Systems. Available at: https://openreview.net/forum?id=0OImBCFsdf.

[6] Wang, Y. _et al._ (2023) ‘Geometric Transformer with Interatomic Positional Encoding’, in _Thirty-seventh Conference on Neural Information Processing Systems_. Available at: https://openreview.net/forum?id=9o6KQrklrE.

**Questions:**

1. As mentioned in W1 authors didn't include recent baselines, some of which even cited in the work. Therefore, authors shown to be aware of those works but why are they decided not to include in the baselines?

2. Authors discussed their computational complexity and mentioned "high-throughput screening" however there is no experiment to support authors claim on the proposed methods' computational complexity compared to the baseline methods.

3. Given that the proposed method utilizing a angular information with $\beta_{ijk}$, how does the complexity remains linear to the number of edges? The clarifications are needed for this since when we consider all possible triplets, the complexity is $n^3$ with respect to the number of nodes $n$.

4. Could the authors provide additional experiments on larger molecular systems or different types of chemical structures to demonstrate the method's generalizability beyond small molecules?

---

### Official Review · Reviewer_RKLt · 2024-11-04

**Soundness:** 1
**Presentation:** 3
**Contribution:** 2
**Rating:** 3
**Confidence:** 4

**Summary:**

This paper proposes EG-SAT, an equivariant graph self-attention transformer for modeling 3D molecular structures, introducing Attention-based Atom-Centered Symmetry Functions (AACSFs) to capture higher-order geometric interactions. The authors showed the performance of proposed model on QM9 and MD17 datasets. While the paper presents some interesting ideas around combining attention mechanisms with ACSFs, there are several critical limitations that need to be addressed.

**Strengths:**

1. The integration of attention mechanisms with ACSFs is interesting.
2. the framework is applicable to multiple molecular property prediction tasks
3. the paper discusses the theoretical foundation of symmetry and group representation in detail

**Weaknesses:**

1. The paper omits several recent works in molecular property prediction, making the comparisons less relevant.
2. The authors didn't conduct ablation studies to evaluate the contribution of different components in the framework.
3. There is no computational efficiency analysis, what's more, the claims of improved scalability are unsupported by any experiments.
4. The motivation for incorporating angular information lacks clear examples where angular information provides benefits.
5. There's no proof that the attention mechanism preserves chemical validity

**Questions:**

1. Can the authors provide experiment for the claimed scalability improvements over traditional ACSFs?
2. How does the computational complexity scale with the number of atoms and chemical elements compared to existing methods?
3. What is the memory footprint of the attention mechanism for larger molecular systems?
4. Can the authors provide ablation studies showing the specific benefits of angular information integration?
5. How sensitive is the model to hyperparameter choices, particularly the attention and gating parameters?

---

### Note · Authors · 2024-11-22

**Comment:**

Thank you for the comprehensive feedback and the time dedicated by the reviewers. Our submission contains several shortcomings and has prompted numerous questions. We recognise that resolving these concerns will need considerable time for revisions in the rebuttal process. Consequently, we have chosen to withdraw our submission for now to enhance the paper's quality.

**Withdrawal Confirmation:**

I have read and agree with the venue's withdrawal policy on behalf of myself and my co-authors.